# Lycopene in the Prevention of Cardiovascular Diseases

**DOI:** 10.3390/ijms23041957

**Published:** 2022-02-10

**Authors:** Sylwia Przybylska, Grzegorz Tokarczyk

**Affiliations:** Department of Fish, Plant and Gastronomy Technology, West Pomeranian University of Technology in Szczecin, 71-459 Szczecin, Poland; grzegorz.tokarczyk@zut.edu.pl

**Keywords:** lycopene, cardiovascular diseases, antioxidant, inflammation, prevention, atherosclerosis, oxidative stress

## Abstract

Cardiovascular diseases (CVDs) are the leading cause of human mortality worldwide. Oxidative stress and inflammation are pathophysiological processes involved in the development of CVD. That is why bioactive food ingredients, including lycopene, are so important in their prevention, which seems to be a compound increasingly promoted in the diet of people with cardiovascular problems. Lycopene present in tomatoes and tomato products is responsible not only for their red color but also for health-promoting properties. It is characterized by a high antioxidant potential, the highest among carotenoid pigments. Mainly for this reason, epidemiological studies show a number of favorable properties between the consumption of lycopene in the diet and a reduced risk of cardiovascular disease. While there is also some controversy in research into its protective effects on the cardiovascular system, growing evidence supports its beneficial role for the heart, endothelium, blood vessels, and health. The mechanisms of action of lycopene are now being discovered and may explain some of the contradictions observed in the literature. This review aims to present the current knowledge in recent years on the preventive role of lycopene cardiovascular disorders.

## 1. Introduction

Cardiovascular diseases (CVD) are a leading social problem worldwide. As a result, not only is the incidence of these diseases high each year, but also the mortality rate. According to statistical data, the number of deaths due to CVD was 17.8 million in 2019, while by 2030, it is predicted to increase to 23 million [1,2]. It is also noted that young people are increasingly afflicted by cardiovascular disease and that one-third of deaths occur in the population under the age of 70. Their main causes are myocardial infarction and stroke [3]. According to the World Health Organization (WHO), cardiovascular diseases include both heart and blood vessel disorders, including hypertension, coronary artery disease, peripheral arterial disease, and heart failure [4]. Regardless of the region of the world, mortality due to cardiovascular diseases is determined primarily by disorders caused by atherosclerosis. This is considered to be a degenerative process of the artery wall caused by oxidative stress, which is manifested by chronic inflammation [5]. In the case of other cardiovascular diseases such as heart failure, key factors contributing to development have a complex relationship between metabolic and molecular changes in oxidative stress, inflammation, lipid metabolism, and endothelial and myocardial dysfunction [6].

Many factors influence the occurrence of cardiovascular disease. According to the American Society for Preventive Cardiology (ASPC) 2021, it is unhealthy nutrition, physical inactivity, dyslipidemia, hyperglycemia, high blood pressure, obesity, considerations of selected populations (older age, race, ethnicity, gender), thrombosis, smoking, dysfunctional kidney disease and genetic/familial hypercholesterolemia [7]. Among them, a significant impact on the risk of CVD is exerted by a particularly unhealthy diet with excessive consumption of SFA, sugar, and salt, and low in plant-derived products, especially fresh fruit and vegetables [8]. Therefore, the Mediterranean diet is one of the main factors that can protect and prevent CVD [9]. The Mediterranean diet (MedDiet), which is based on minimally processed food, rich in fruit and vegetables, including tomatoes and tomato-based products and olive oil (EVOO), seems to be an ideal nutritional model for maintaining a healthy circulatory system [10]. Tomatoes, which are an integral part of the Mediterranean diet and are eaten fresh or processed as juices, concentrates, ketchups, or soups, also provide valuable bioactive ingredients for the body. These include, among others: carotenoids (lycopene, β-carotene), phenolic compounds (quercetin, kaempferol, naringenin), and vitamins (ascorbic acid, tocopherol) [11]. Due to this, tomato products are of great interest to scientists whose research work focuses on searching for a relationship between lycopene consumption and the occurrence of CVD [12]. Their numerous epidemiological studies provide timely information on lycopene, confirming its importance in the prevention of cardiovascular diseases [13]. Lycopene is believed to have a protective effect not only on the cardiovascular system but also to have potential benefits in preventing cancer. It is highly effective against prostate, stomach, lung, and breast cancer. More importantly, this compound, apart from suppressing the proliferation of neoplastic cells, induces their programmed death and inhibits metastasis. Lycopene also has a high pro-health activity on the skeletal system. It improves bone mineralization and protects against the occurrence of osteoporosis. In addition, it has neuroprotective properties, which makes it helpful in the prevention and treatment of neurodegenerative diseases, such as Alzheimer’s disease, Parkinson’s disease, or Huntington’s disease [14].

Lycopene is a naturally occurring red pigment in tomatoes, watermelon, pink grapefruit, papaya, apricots, and guava [15]. The main sources of lycopene in the diet are tomatoes and tomato-based products (Table 1) [16,17].

Whereas tomato pomace, on the other hand, is a promising raw material for the production of functional foods [18]. More than 80% of dietary lycopene intake in the developed world is derived from processed tomato products such as ketchup, tomato juice, spaghetti sauce, and pizza sauce [19]. Tomato products from a dietary point of view are a source of lycopene with increased bioavailability due to thermal treatment, but also due to the fact that the processing releases lycopene from the fibrous matrix of the cellular structure [20]. Regardless of the individual conditions of the European population, the average consumption of lycopene ranges from 5 to 7 mg/day [21]. In the United States, its consumption among men ranges from 6.6 to 10.5 mg/day and among women from 5.7 to 10.4 mg/day [22].

Taking into account the fact that humans are not able to synthesize lycopene de novo, therefore its supply with food is necessary to take advantage of its pro-health properties. Currently, more and more clinical trials see many significant health benefits in lycopene, including its protective effect on cardiovascular diseases [23]. In addition to being a carotenoid with the highest antioxidant potential, it also has the ability to modulate inflammation, apoptosis, and cellular communication, which is important in the context of cardiovascular diseases [24]. Moreover, a diet rich in antioxidants could be promising for preventing and treating COVID-19, but strong clinical research data are required to support this claim [25]. The pandemic has increased consumer concerns about overall health and has generated opportunities for innovative functional foods containing target bioactive compounds, such as lycopene [26,27].

Lycopene, of the 24 identified carotenoids in human plasma, has the highest level. It has been found, however, that its amount is largely influenced by the diet and the share of products rich in this compound in it. It is assumed that its average content in the population plasma, depending on the geographic region, may range from 0.11 μM (in the Japanese population) to 1.32 μM (in the Italian and Greek populations) [28]. On the other hand, its plasma/serum half-life is quite different and, according to some sources, it ranges from 2 to 3 days, and to others from 12 to 33 days [29].

In fresh, ripe tomatoes, the *all-trans* lycopene isoform is the dominant form and accounts for about 90%. In turn, their processed products, such as ketchups, tomato concentrates, juices, are dominated by the forms of *cis*-lycopene. It is assumed that thanks to the *cis* form, lycopene is highly bioavailable in the human diet [30,31]. Moreover, some studies confirm that increasing the bioavailability of lycopene as a result of its *trans* to *cis* isomerization can be achieved by adding fish oil or olive oil to tomato dishes. As a result of the interaction of compounds from other products with lycopene, including sulfur compounds derived from onion or garlic, lycopene isomerized, and thus its bioavailability in the form of *cis* increases [32].

## 2. Mechanisms of Lycopene Action in the Cardiovascular System: Antioxidant Effect and Anti-Inflammatory Effect

### 2.1. Antioxidant Effects of Lycopene

Increased oxidative stress is considered to be the primary factor in cardiovascular disease. It causes, among others, myocardial infarction, ischemia/reperfusion, and heart failure [33]. The production of excess ROS contributes to the reduction of nitric oxide availability and vasoconstriction, initiating arterial hypertension. ROS also has a negative effect on the calcium management in the myocardium, causing arrhythmia and enhancing the remodeling of the heart by inducing signal hypertrophy and apoptosis. More importantly, ROS also initiates the formation of atherosclerotic plaques [34]. Therefore, bioactive dietary compounds, including lycopene, may be important in counteracting these changes. Lycopene is a carotenoid pigment with a molecular structure of C_40_H_56_ (Figure 1) [35,36].

It is a straight-chain highly unsaturated hydrocarbon that has a total of 13 double bonds, 11 of which are conjugated bonds. Due to this structure, it is considered to be one of the strongest antioxidants in food [37]. Lycopene is a highly effective antioxidant that, due to the high reactivity between the long polyene chain and free radicals, enables the elimination of singlet oxygen and the reduction of reactive oxygen species (ROS) [38]. The reactivity of lycopene, in biological systems depends on their molecular and physical structure, location or site of action within the cells, ability to interact with other antioxidants, concentration, and the partial pressure of oxygen [39]. Biologically, lycopene tends to act as singlet oxygen (1O_2_) and peroxyl radical scavenger (LOO•) [40]. The mode of action of lycopene towards reactive forms of ROS can be predicted through three mechanisms such as (Figure 2):

Besides quenching singlet molecular oxygen, lycopene is also known to act on other free radicals like hydrogen peroxide, nitrogen dioxide, and hydroxyl radicals [42]. Lycopene is believed to increase the cellular antioxidant defense system by regenerating non-enzymatic antioxidants, such as vitamins E and C, from their radicals [43].

It is believed that the cardioprotective effect of lycopene protection is a result of its potential antioxidant properties responsible, inter alia, for: protection against oxidative stress-induced myocardial hypertrophy by improving ROS production [44], inhibition of stress-induced endoplasmic reticulum damage due to ischemia/reperfusion (I/R) [45], inhibition of LDL oxidative damage [46]; suppression of ventricular remodeling after myocardial infarction by inhibiting apoptosis [47], and improving endothelial function [48].

Lycopene is considered an effective singlet oxygen quencher in the carotenoids group [49]. It is a much more potent antioxidant than alpha-tocopherol (10× more potent) or beta-carotene (twice as potent) [50]. The basic mechanisms of the antioxidant role of lycopene include reducing the growth of ROS and inhibiting oxidative stress. In addition, suppression of inflammatory (TNF-α, IL-6 and IL-1β), NF-kB, and apoptotic (caspase and Bbl-2) pathways. By activating the antioxidant response element (ARE) associated with nuclear factor E2 (NFE2L2), it increases the amount of antioxidant enzymes, which include superoxide dismutase (SOD), catalase (CAT), and glutathione peroxidase (GSH-Px) [51]. Due to its antioxidant properties, lycopene is also important in the protection of DNA, lipids, and other macromolecules involved in the proper functioning of the body [52] (Figure 3).

Oxidative stress is believed to lead to cardiac hypertrophy. Lycopene, on the other hand, can inhibit cardiac hypertrophy by alleviating oxidative stress. This is confirmed, among others, by the in vitro and in vivo studies conducted in 2019 by Zeng et al., who showed that lycopene can inhibit cardiac hypertrophy caused by pressure overload [44]. The authors found that it could reverse the increase in reactive oxygen species (ROS) production during the process of hypertrophy and slow down the activation of ROS-dependent pro-hypertrophic MAPK and Akt signaling pathways. Moreover, they noticed that lycopene activated the expression of antioxidant genes induced by ARE [53]. The hypertrophy of the heart muscle itself (hypertrophy) is associated with an increase in the size of the cardiomyocytes. As a result, the size of the heart increases, most often its left ventricle. Additionally, its hypertrophy may be a consequence of increased incidents of atrial fibrillation, arrhythmias, and sudden cardiac death [54]. It is assumed that the hypertrophy of the left ventricle is a consequence of the proliferation of fibroblasts and myocytes. Fibroblast proliferation is mediated by substances such as angiotensin II, endothelin-1, and aldosterone. In contrast, myocyte proliferation is mediated by growth hormone and thyroxine. For example, in studies on cardiomyocyte cultures, their overgrowth caused by phenylephrine (PE) and angiotensin II (Ang II) increased the increased production of ROS, the activity of which was in turn suppressed by antioxidants, including lycopene [55,56]. Moreover, increased ROS production may modify the activation of mitogen-induced protein kinases (MAPK), Akt, and calcineurin, which are believed to be the cause of cardiac hypertrophy [53]. According to an in vitro study conducted in 2014 by Chao et al., it was found that lycopene was effective in suppressing the hypertrophy of cardiac cardiomyocytes caused by the use of urotensin II [57]. In addition to this property, lycopene in this study (used in an amount from 3–10 μM) reduced the reactive forms (ROS) initiated by urotensin II and decreased the level of NAD (P) H oxidase-4 expression. In addition, lycopene inhibited the increase in urotensin II-stimulated phosphorylation, Akt serine-threonine kinase, and glycogen synthase-3beta kinase (GSK-3β). On the other hand, on the basis of the research conducted in 2015 by Hong et al., it was found that lycopene (in the form of watermelon powder) reduced inflammation by reducing the activity of the pro-inflammatory mediator cyclooxygenase 2 (COX-2), disrupting the production of prostaglandins E2 and I2. As a consequence, it reduced cardiovascular disorders [58]. According to the authors, the use of watermelon powder containing lycopene in the diet is a beneficial supplementation, because it has a high anti-inflammatory effect similar to COX-2 inhibitors and conventional non-steroidal anti-inflammatory drugs.

### 2.2. Anti-Inflammatory Mechanism of Lycopene

According to scientific research, inflammation corresponds to and initiates a number of diseases related to the cardiovascular system as well as many others, including hypercholesterolemia, type 2 diabetes, and obesity [59]. Inflammation is considered to be the main cause of coronary heart disease (CHD), which is based on its atherosclerotic changes. The conducted translational and preclinical studies on humans confirm that the presence of inflammation in their bodies is responsible for the development and severity of atherosclerosis. On the other hand, epidemiological studies assume that intense and long-lasting systemic inflammation may be a consequence of cardiovascular diseases. A 2016 study by Teague and Metha confirms that the residual risk of inflammation (RIR) increases the severity of coronary heart disease by modulating both immune cells and inflammatory cytokines [60]. Lycopene is of great importance in inhibiting the inflammatory response, and its activity in this direction is mainly related to the suppression of its basic mediators, such as reactive oxygen species (ROS). Moreover, this compound inhibits the synthesis and release of pro-inflammatory cytokines, including IL-1β, IL-6, IL-8, and TNF-α. Lycopene’s anti-inflammatory activity is thought to be mediated at multiple levels, primarily through inhibition of the nuclear factor κB (NF-κB), regulation of mitogen-activated protein kinase (MAPK), inducible nitric oxide synthase (iNOS), and the inhibition of enzymes involved in the metabolism of arachidonic acid—cyclooxygenase-2 (COX-2) and lipoxygenase (LOX) [61] (Figure 4).

According to the research of Hung et al., 2008, lycopene has the ability to suppress TNF-alpha-induced activation of NF-kappa B, reduce the expression of intracellular adhesion molecule-1 (ICAM-1) and interactions between monocytes and endothelial cells, thus emphasizing its high vascular efficacy [62]. Whereas, the studies by He et al., carried out in 2016, show that lycopene reduces the risk of graft vasculopathy. The effect of this is its high activity in the suppression of intimal hypertrophy and smooth muscle cell proliferation and diminution of inflammation in the vessels of the allograft [63]. According to these authors, lycopene appears to be of potential importance in the suppression of allograft atherosclerosis by lowering Rho-coupled kinases and regulating the expression of key factors via the NO/cGMP pathways. An increased inflammatory response and cardiomyocyte apoptosis characterized by inflammation, apoptosis, infarction, and fibrosis are key processes in ventricular remodeling after MI. On the other hand, the administration of lycopene in the amount of 10 mg/kg body weight in mice resulted in inhibition of the NF-κB signaling pathway. The result was a reduction in the expression of fibrosis mRNA (TGF-β1, collagen I, collagen III), inflammatory markers (TNF-α, IL-1β), and markers of apoptosis (caspase-3, -8 and -9), which, consequently, reduced the inflammatory response and cardiomyocyte apoptosis after MI [45]. In contrast, the studies by Le et al. showed that the use of lycopene in the diet for 21 days in the amount of 5 mg/kg enhances the expression of transforming growth factor beta 1 (TGF-β1), which results in protection against changes caused by atrazine (ATR—synthetic herbicide) and inflammation of the heart [64]. The above studies showed that the effectiveness of lycopene was also associated with the reduction of the amount of NO increased by atrazine and the activity of NOS. In addition, lycopene suppressed atrazine-induced levels of the proinflammatory cytokines TNF-α, IL-6, IL-1β, and COX-2 and the activation of NF-κB, as confirmed by its anti-inflammatory effect in the protection of the myocardial. In the case of studies on the effect of lycopene on changes induced by myocardial infarction (MI) in a rat model, it was noticed that it suppressed the increase in MMP-9 and type I collagen expression, and inhibited p38 activation and decreased collagen levels in the periinfarct zone [65]. In the studies of Ferreira-Santoset et al., the use of 10 mg/kg of lycopene inhibited the angiotensin II-induced changes in the cardiovascular system. Lycopene at this level reduced myocardial hypertrophy and its fibrosis and showed a beneficial effect in reducing arterial hypertension [66]. In 2017, Yang et al. showed that lycopene inhibits TNFα-induced adhesion of monocytes to EC (endothelial cells) by reducing ICAM-1 expression [67]. The studies also confirmed that lycopene suppresses activation of the inflammatory transcription factor NF-κB by blocking the degradation of IκBα. The transcription factor NF-κB is believed to be responsible for a number of inflammatory diseases in the human body, including atherosclerosis, and its activation by TNFα is required for transcriptional activation of EC adhesion molecules [68]. This confirms that lycopene inhibits the expression of adhesion molecules by blocking NF-κB activation. Pretreatment with lycopene for 3 h, according to studies by Yang et al., suppresses TNFα-induced IκBα degradation [67]. Blocking NF-κB activity by lycopene is assumed to be mediated by modulation of upstream targets in the NF-κB pathway. Moreover, a short incubation with lycopene for 3 h only partially inhibited the appearance of NF-κB in EC nuclei. On the other hand, extending it to 12 h increased the effect of lycopene on p65 translocation [67]. The research carried out in 2015 by Sung et al., showed that lycopene can enhance the activation of the phosphoinositide 3-kinase/Akt pathway, and then the induction of Nrf2 nuclear translocation in the EC [69]. According to Yang et al., lycopene also induces the expression of the phase II enzyme GCL and HO-1, and Nrf2 translocates to the nucleus and binds to ARE to activate transcription. The anti-inflammatory effect of lycopene is mediated by inhibition of IκBα degradation, and ICAM-1 expression is reduced by siRNA transfection of HO-1 [67]. Increased expression of HO-1 in atherosclerotic plaques with the participation of lycopene suppresses the progression of atherosclerotic disease, which confirms its protective role. In addition, the anti-inflammatory effect of lycopene-induced HO-1 was investigated by Sahin et al. [70].

## 3. Efficacy of Lycopene in Inhibiting the Risk of Cardiovascular Diseases

In 2020, Tierney et al., described in their review 43 studies related to the assessment of the impact of lycopene supplied from food and its supplements on the risk of cardiovascular disease. The share of lycopene in the diet of the respondents depending on the form of its administration (supplements with or without food, based on tomato juice/paste /raw product or in combination with olive oil) was quite diverse and its daily dose ranged from 1.44 to 75 mg. In 11 out of 43 studies, lycopene did not reduce risk factors for cardiovascular disease [71]. It is believed that in addition to the strong antioxidant activity with potential health-promoting properties of lycopene, the cardiovascular system is also influenced by its anti-inflammatory, anti-atherosclerotic, and anti-platelet effects. Moreover, it also improves endothelial function (EF; blood flow and NO bioavailability). Lycopene can bind to plasma LDL cholesterol and by this mechanism, it provides protection against atherosclerosis by suppressing lipid peroxidation.

### 3.1. Atherosclerosis and the Action of Lycopene

High blood lipids, including cholesterol and triglycerides (TG), constitute the primary risk of cardiovascular disease (ASCVD) [72]. It is assumed that atherosclerosis is the main factor responsible for the occurrence of most heart disease entities, including acute coronary syndromes. Atherosclerosis, also known as ischemic heart disease, most often affects people over 45 years of age. It is most often caused by an unhealthy diet rich in animal fats, stimulants, obesity, and diabetes. The cause of atherosclerosis is the narrowing of the lumen of the coronary arteries, which significantly impedes the free flow of blood to the heart muscle. Accumulation of lipids on arterial walls causes the formation of atherosclerotic plaques, and elevated ROS levels induce endothelial dysfunction, vasculitis, and accelerated accumulation of low-density lipoprotein (ox-LDL) in the arterial wall, which enhances the progression of atherosclerotic lesions of the heart [73]. Both increased levels of ox-LDL and ROS can damage mitochondria, damage mitochondria, release mi-ROS, induce NLRP3 activation, raise IL-1β and IL-18 levels, and cause inflammation [74,75]. Moreover, the presence of increased ROS production leads to the degradation of endothelial nitric oxide synthase (eNOS) by increasing the activity of mitochondrial arginase II [76] (Figure 5).

It is assumed that controlling cardiovascular risk factors such as a healthy diet, obesity treatment, and lowering high blood sugar levels can significantly reduce mitochondrial stress and reduce their damage [77]. More importantly, the elimination of the so-called junk foods that are high in trans fatty acids (TFA), TG, LDL, and low in HDL can significantly improve patients’ lipid profiles. On the other hand, an increased level of TFA additionally increases the amount of pro-inflammatory cytokines, causing endothelial dysfunction and insulin resistance [78]. Artificial TFAs are associated with an increased risk of atherosclerosis and cardiovascular events [79].

Therefore, it is assumed that the normalization of lipid metabolism by reducing total serum cholesterol, triglycerides, and the LDL-cholesterol fraction is the main therapeutic goal in reducing the risk of cardiovascular diseases. Currently, one of the prophylactic agents that lower LDL-C is statins, inhibitors of HMG-CoA reductase (HMGR)—the rate-limiting enzyme in the cholesterol biosynthesis pathway. It is assumed that their action to lower LDL-C levels is mediated by modulation of SREBPs, which increase LDL-R expression in the liver, which results in increased LDL-C clearance from the circulation [80,81]. It should be noted, however, that statins may not be effective in lowering lipids and reducing the risk of ASCVD in all patients. Some of them may show increased intolerance to them, which in clinical practice is quite a serious problem [82]. Taking this into account, however, it should be emphasized that the use of statins is able to reduce the incidence of ASCVD by 70% [83]. It should be noted that recently the subtilisin/kexin type 9 proprotein protein convertase (PCSK-9) has occupied an important place in cardiovascular medicine. PCSK-9 belongs to the proprotein convertase (PC) family of serine proteases that has reflective effects on plasma LDL-C levels in response to its ability to direct cell surface LDL-R protein to the lysosomes for degradation, resulting in reduced and altered clearance of LDL-C and subsequent accumulation in the circulation [84]. In 2017, Alvi et al. showed that lycopene can lower hypercholesterolemia by targeting the expression of the liver genes PCSK-9 and HMGR, and by reducing the affinity of PCSK-9 to form a complex with EGF-A-like LDL-R repeats, leading to an increase in LDL-R activity and the subsequent elimination of LDL-C from the body [85]. Observations obtained indicate that supplementation with lycopene in this form may be particularly beneficial for patients intolerant to statins. To emphasize, lycopene, unlike these and other synthetic drugs, is a natural product. In addition, it has the advantage that even despite high doses, it is not toxic, completely safe, not harmful to the environment, and cheap. Therefore, it is highly effective in inhibiting hypercholesterolemia and thus reducing maybe the occurrence of ASCVD [86,87].

Lycopene is not able to increase HDL cholesterol, but was shown to improve the LDL/HDL ratio, HDL functionality and reduced the accumulation of cholesterol in the rabbit aorta, underlining the beneficial effects of lycopene during the initial stages of atherosclerosis [24].

Lycopene has been found to lower the synthesis of dysfunctional HDL by modulating HDL functionality towards an anti-atherosclerotic phenotype, with low serum amyloid A levels and favorable changes in HDL remodeling enzymes (cholesterol ester transfer protein and lecithin cholesterol acyltransferase) [88]. A hypotriglyceridemic effect of tomato juice was seen only in subjects with initial high serum triglyceride levels [89].

In the disease of diabetes, both hyperglycemia, insulin resistance, and dyslipidemia intensify inflammation and pro-oxidative conditions, accelerating the development of atherosclerosis. Moreover, in this case, atherosclerosis begins earlier, is more extensive, and progresses more strongly than in people without diabetes [90]. In 2020, Chiva-Blanch et al. investigated the potential relationships between plasma lycopene isomers and atherosclerosis in people with diabetes [91]. Their plasma lycopene levels were inversely related to the burden of atherosclerosis, indicating that eating tomatoes and tomato by-products, in the context of a healthy diet, may be beneficial in suppressing atherosclerosis and, consequently, heart disease. According to the authors, the plaque load was inversely proportional to 5-cis-lycopene, all cis-lycopene isomers, trans-lycopene, and total lycopene isomers (all, *p* < 0.05). In 2019, Nishimura et al., in a clinical study, evaluated the effect of daily consumption of tomato with a high content of lycopene for 12 weeks on lipid profiles in Japanese patients with LDL-C ≥120 mg/dL and <160 mg/dL [92]. Consumption of semi-dried tomatoes with high lycopene content (PR-7) increased their plasma lycopene levels and improved lipid profile compared to the placebo group. In 2017, Kumar et al. demonstrated that lycopene had a positive effect on the reduction of serum TC, LDL-C, VLDL-C, and triglycerides, and increased HDL-C values compared to rats from the control group. This showed that lycopene had a beneficial effect on hyperlipidemia [93]. According to the authors, this property could be due to the prevention of lipid oxidation by lycopene due to its high antioxidant activity. Moreover, lycopene had a lower TBARS value which can be attributed to its antioxidant properties and non-oxidative mechanisms, such as inhibition of 3-hydroxy-3-methyl-glutaryl-coenzyme A reductase, suppression of endothelial tissue cell factor, and inhibition of expression of cell adhesion molecules.

Lycopene also plays an important role in inhibiting cardiovascular complications, including blood clots resulting from atherosclerotic changes. Based on in vitro and in vivo studies, the anti-atherosclerotic activity of lycopene is mainly due to: inhibition of endothelial damage, suppression of cholesterol synthesis by inhibition of 3-hydroxy-3-methylglutaryl-coenzyme A reductase, inhibition of LDL oxidation and restoration of HDL functionality, lowering the level of TG and LDL-cholesterol, inhibition of pro-inflammatory activity directed by macrophages and T lymphocytes [24,94].

The anti-atherosclerotic properties of lycopene are also associated with inhibition of vascular smooth muscle cell (VSMC) proliferation and foam cell formation [95].

### 3.2. Hypertension and the Effect of Lycopene

Hypertension is one of the main factors that increase the risk of cardiovascular diseases, including coronary heart disease, stroke, and atrial fibrillation. The presence of this disease entity contributes most to heart failure [96,97]. In 2017, the AHA/ACC defined hypertension as systolic blood pressure (BP) greater or equal to 130 mmHg, or a diastolic BP of 80 mmHg or greater [98]. Hypertension loads the heart muscle as a result of its functional and structural changes. These changes are mainly associated with left ventricular hypertrophy, which can lead to heart failure. Patients with left ventricular hypertrophy have significantly increased morbidity and mortality [99]. Hypertension is one of the most common pathologies in America, affecting approximately 75 million adults, or one in three adults. Of those patients diagnosed with hypertension, only 54% have adequate blood pressure control [72]. The global prevalence of hypertension is 26.4%, which is 1.1 billion people, but only one in five people is adequately controlled. Despite the availability of effective antihypertensive drugs, the achievement of target blood pressure (BP) is still low worldwide [100]. Among treated patients, only 36% achieve an improvement in BP [101]. Although antihypertensive drugs are safe and effective, they can nevertheless induce adverse reactions such as electrolyte disturbances, acute kidney damage, ankle swelling, and hypotension due to excessive BP reduction [102].

Introducing a healthy lifestyle may seem to be an effective method in the prevention and treatment of arterial hypertension [103]. Many studies show that healthy eating habits, and especially high dietary intake of fruit and vegetables and supplements with antioxidant properties, are able to guarantee the improvement of vascular function and BP [104]. Due to this, supplementation with bioactive food ingredients, such as lycopene, in the control of arterial hypertension is important in this case. Lycopene has an antihypertensive effect by inhibiting the angiotensin converting enzyme (ACE) and due to its antioxidant effect, reducing the oxidative stress induced by angiotensin II and indirectly increasing the production of nitric oxide in the endothelium [105] (Figure 6).

An example is a study in which 8556 overweight and obese adult participants showed reduced hypertension after increasing the daily diet of lycopene [106]. There have been many clinical studies investigating the relationship between lycopene supplementation and its effects on blood pressure. However, the results obtained were different. Some suggest that lycopene supplementation had a beneficial effect on blood pressure [107,108,109]. On the other hand, there are some studies in which no effect on changes in blood pressure was found [110,111]. Surprisingly, one study even found that lycopene supplementation could raise BP [112]. In contrast, a recent study found that treatment with a tomato nutrient complex (TNC) standardized to 15 or 30 mg of lycopene was associated with a significant reduction in systolic BP (SBP) [12].

Most epidemiological studies emphasize the beneficial role of tomatoes and tomato products in the improvement of vascular function, which is mainly related to its high antioxidant activity, which has a good effect in reducing arterial hypertension [113]. The antihypertensive mechanism of lycopene is believed to involve inhibition of the angiotensin converting enzyme (ACE), thereby reducing the oxidative stress induced by angiotensin II and indirectly increasing the production of nitric oxide (NO) [13]. Oxidative stress, by disrupting NO production, leads to oxidative damage to endothelial cells and ultimately to endothelial dysfunction [114]. Lycopene, by enhancing the activity of antioxidant enzymes, such as catalase and superoxide dismutase, protects endothelial cells against oxidative damage and regulates blood pressure [48]. Scientific research indicates that adiponectin plays an important role in blood pressure regulation, which it lowers through anti-inflammatory and anti-atherosclerotic effects and reverses salt-induced hypertension in the diet [115]. The latest results of a meta-analysis conducted in 2021 by Kelishadi et al., showed that lycopene supplementation is more effective in improving (systolic blood pressure) SBP when patients receive higher doses of it for longer periods of time. According to the results of this study, the observed decrease in SBP was more significant with lycopene supplementation at a dose of ≥15 mg/day [116]. In 2013, Li et al., in a meta-analysis conducted, found that high doses of tomato extract (>12 mg/day) were the most effective in lowering BP levels compared to lower levels [117].

## 4. Lycopene Supplementation and the Risk of Cardiovascular Diseases

It should be noted that even well-planned nutritional concepts, the role of which is to enrich the human body with lycopene, may not be fully sufficient to compensate for this deficit. The result is its poor intestinal absorption, low bioavailability, and decreased liver capacity by incorporating lycopene into lipoprotein carriers [118]. Therefore, the development of new lycopene supplements with increased bioavailability is of great importance in the light of modern nutritional science. It is assumed that this will allow you to take full advantage of its health-promoting properties in order to protect the body against various diseases, including cardiovascular diseases. Taking into account such assumptions, it can be concluded that the application of, inter alia, emulsions may show promise in creating nutraceuticals with enhanced bioavailability in tomato-based products. An example is the development of such products in 2021 by Nemli et al., in which the presence of excipient emulsions significantly increased the bioavailability of lycopene in tomato pomace compared to oil-free controls [119]. By comparison, the bioavailability of lycopene was >82% when tomato pomace was mixed with excipient emulsions, but only 29% when mixed with oil-free buffer solutions.

In their 2018 research, Petyaev et al. showed that the consumption of lycosome-formulated lycopene increased its serum level by 2.9 and 4.3 times, respectively, after 2 and 4 weeks of use; while lactolycopene supplementation increased its level, it doubled only after 4 weeks [120]. The use of a lycosome preparation containing lycopene in the present study suppressed inflammation by reducing the marker of oxidative damage and a fivefold reduction in the level of oxidized LDL. At the same time, supplementation with lycopene in this form as a lycosome was the result of better tissue oxygenation. For comparison, lactolycopene did not cause significant changes in the parameters tested. On the other hand, the improved bioavailability of lycopene in the form of a lycosome increased its antioxidant and anti-inflammatory properties and had a positive effect on the cardiovascular system of the analyzed population. According to a study by Müller et al., lycopene supplementation is definitely more beneficial for people with a deficiency of antioxidants, such as the elderly and those exposed to higher levels of oxidative stress, such as smokers, diabetics, or after a myocardial infarction [121]. Quite interesting results in their research on the health-promoting properties of lycopene were obtained by Thies et al., in 2017. Based on the collected data from 54 intervention studies, the authors concluded that the use of various lycopene supplementations in most of the analyzed studies shows greater effectiveness in the treatment of neoplastic diseases and only 13 in cardiovascular diseases. The main effectiveness of lycopene towards the protective effect on the cardiovascular system was related to the reduction of C-reactive protein, improvement of lipid profiles, as well as reduction of blood pressure, and inhibition of pro-inflammatory mediators [24]. Taking into account the physicochemical properties of lycopene, it should be noted that its pro-health effectiveness depends on the form and factors that facilitate its absorption and assimilation from the diet [122]. It is assumed that the degree of its absorption by the human body is influenced by the form and type of the consumed product (fresh, processed), the presence of fat, or its more improved form using the modern method of nanotechnology. In addition, the use of lycopene is also determined by the individual characteristics of the population itself, such as age, sex, race, and lifestyle [123].

The bioavailability of lycopene is influenced by several factors, such as the season of the year, the way tomatoes are processed, their origin, size, and form of consumption [124]. Its isomerization is also important. Fresh tomatoes contain lycopene in *all-trans* form. Several factors, including high temperatures, light, oxygen, acids, and metal ions (Cu, Fe) cause the isomerization of lycopene [18]. Under the influence of technological processes such as cooking, pasteurization, or drying, the natural form of trans lycopene is destroyed into the cis form. It is assumed that dried tomato products, such as tomato powder, have the highest level of cis-lycopene forms. Due to what is confirmed by the research conducted by Colle et al. in 2013, despite the fact that powdered tomatoes show reduced stability of lycopene, thanks to the high content of it in the form of cis, they are the best digestible form for the human body [125]. They are also highly effective in integrating into lipoproteins, thanks to which they can be transported and absorbed to the body’s target sites, ensuring effective protection against cardiovascular diseases. According to Friedman [126], eating lycopene-containing foods is more effective than using lycopene supplements. The result is the presence of other bioactive compounds in tomato products that may increase the bioavailability of lycopene. In a study conducted with heart failure patients, lycopene consumption promoted a decrease in C-reactive protein levels in women but had no significant effect in men [127]. In another study, lycopene supplementation was able to improve vasodilation mediated by endothelial cells, in cardiovascular disease patients, while in the healthy control patients, no significant effect was observed. It is generally accepted that consuming one or more servings of lycopene-rich tomato products per day is associated with less than a 30% risk of cardiovascular disease [128]. According to Cheng et al., the consumption of tomato products in amounts (70–400 g/day) for 1–180 days, provides high cardioprotective protection [129].

The following Table 2 presents examples of clinical trials and meta-analyzes on healthy, obese, and sick people.

The use of lycopene in the form of various doses in animal studies shows its positive effect in protection against cardiovascular diseases (Table 3).

From a safety point of view, lycopene seems to be a completely safe component of the diet. Its use in an optimal dose of 12 mg/day to a very high 150 mg/100g does not show any toxic effect in the populations assessed in the literature [141]. Moreover, in animal studies such as (rats, dogs, rabbits) and in vitro, which concerned the assessment of acute toxicity, genotoxicity, and metabolism of lycopene, the negative effects of synthetic and natural lycopene (up to 3 g/body weight/day) were not observed [142]. For example, Wistar rats showed no adverse effects of various doses of natural lycopene supplementation (0–616 mg/kg body weight/day with Blakeslea trispora for 90 days) on neurobehavioral observations, motor activity, body weight, or other clinical and physical parameters [142]. Based on evidence from human clinical trials, Eliassen [143] proposed 75 mg per day as the upper level of supplements (ULS). Based on the No Observed Effect Levels (NOAEL) for synthetic lycopene formulation in various toxicity studies, the European Food Safety Authority (EFSA) panel established the Acceptable Daily Intake (ADI) at 0.5 mg/kg body weight/day [144].

## 5. Conclusions and Future Perspectives

The present review supports the importance of lycopene in improving vascular function and in the prevention of cardiovascular disorders.

Lycopene is one of the most widely investigated carotenoid compounds not only in neoplastic diseases but also in cardiovascular diseases. In our review, we discussed the most important aspects of lycopene in the prevention of cardiovascular diseases, with particular emphasis on atherosclerosis and hypertension. As shown by epidemiological data, their occurrence is the main factor of cardiovascular disorders and related deaths all over the world. Therefore, in order to prevent this, it is necessary from an early age to take care not only of a healthy lifestyle but especially a healthy diet. It is important to include as many vegetables and fruits as possible in your diet. Tomatoes rich in bioactive ingredients, including lycopene, may be of particular importance here. Both the consumption of lycopene in the form of fresh tomatoes and their products as well as in supplements can bring positive effects in reducing the occurrence of cardiovascular diseases. A large number of epidemiological studies have advocated that daily intake of 2–20 mg lycopene has significant benefits in the prevention of CVD. Introducing lycopene into the diet and improving cardiovascular health is mainly related to its strong antioxidant and anti-inflammatory properties. The discussed experiments confirmed that lycopene is an effective antioxidant, the supply of which in the diet can improve blood pressure, endothelial function, metabolic profile, as well as reduce the size of atherosclerotic plaque.

Despite some contradictory studies as to the effects of lycopene on CVD diseases, a growing body of evidence shows undoubtedly its benefits in improving the health of people, among others, with atherosclerosis and hypertension. It should be noted that the mechanisms underlying the action of lycopene are still being discovered and may explain some of the contradictions observed in the literature. Interindividual genetic differences can be a key issue in understanding the real effects of lycopene in each individual. The form of consumption of lycopene also seems to be important, as does the frequency of its consumption in the form of fresh tomatoes or processed products such as sauces, juices, concentrates, ketchups, and as a supplement. Another important aspect regards the bioavailability of dietary lycopene. Lycopene seems to be better absorbed from processed tomato products than from fresh tomatoes. Another important aspect relates to the bioavailability of lycopene in the diet. Lycopene appears to be better absorbed from processed tomato products than from fresh tomatoes. The level of its bioactive properties is influenced by many factors, such as bioavailability, metabolism, isomerization or interactions with other carotenoids.

Lycopene, despite its cardioprotective protection, still has limited biomedical use due to poor bioavailability, which can be solved by advanced systems such as nanotechnology or regular consumption of tomatoes in a processed or fresh form with the addition of olive oil. While many aspects of the metabolism, function, and clinical indications of lycopene in vivo remain to be elucidated, the consumption of tomato-based products can be regarded as an effective preventive measure against cardiovascular disease.

## Figures and Tables

**Figure 1 ijms-23-01957-f001:**
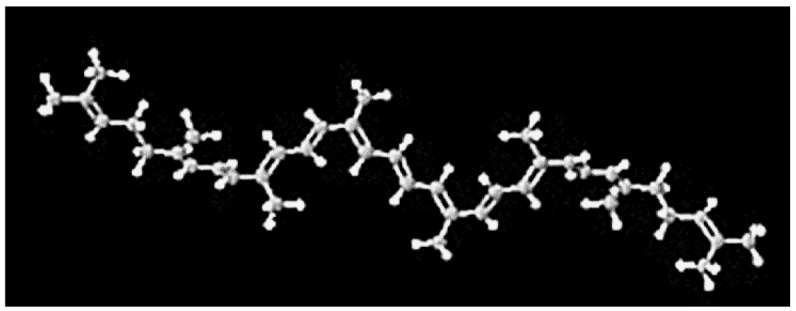
Chemical structure of lycopene [36].

**Figure 2 ijms-23-01957-f002:**
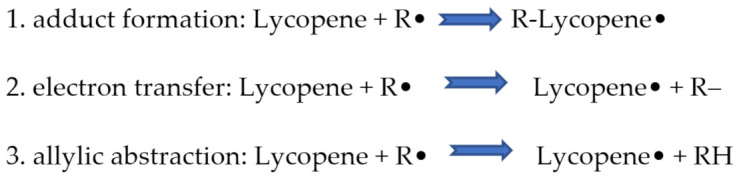
Three possible reactions of lycopene with radical species [41].

**Figure 3 ijms-23-01957-f003:**
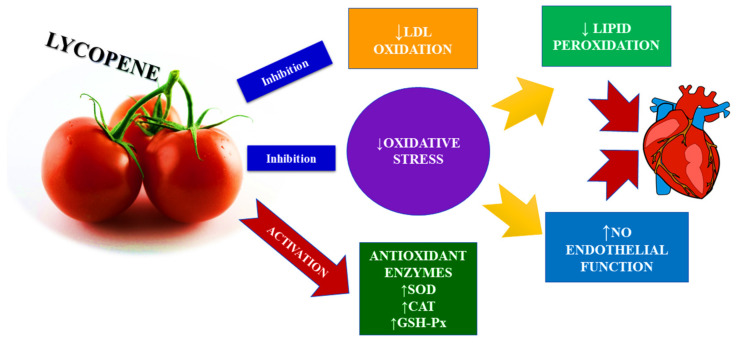
The antioxidant mechanism of lycopene in CVD.

**Figure 4 ijms-23-01957-f004:**
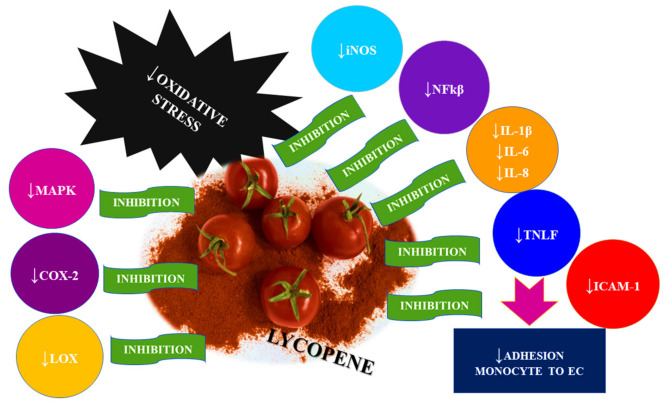
Lycopene and its influence on inflammatory mediators.

**Figure 5 ijms-23-01957-f005:**
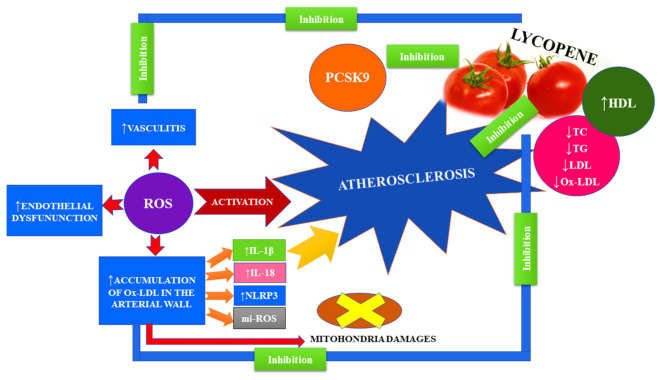
The effect of lycopene atherosclerosis.

**Figure 6 ijms-23-01957-f006:**
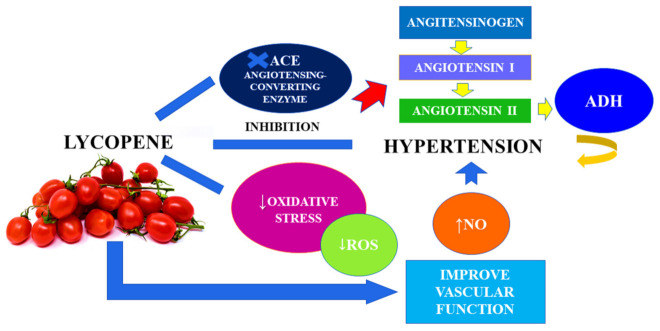
The effect of lycopene on the reduction of blood pressure.

**Table 1 ijms-23-01957-t001:** Lycopene content in tomato and tomato products [16,17].

Source	Lycopene Contents (mg/100 g)
Fresh tomato	5.6
Tomato juice	9.04
Ketchup	16.6
Tomato sauces	23.8
Tomato concentrate	54.0
Tomato puree	21.7
Tomato sun-dried	45.9

**Table 2 ijms-23-01957-t002:** The cardioprotective effect of lycopene on humans in clinical trials and meta-analyzes.

Study Authors/Country	Area of Interest	Study Design	Age and Health Condition	Dose and Treatment Period of Lycopene	Main Findings	Reference
Misra et al., 2006 (India)	Lipid profile	Parallel, RCT	41 healthy postmenopausal women at the age of 46	Two capsules of lycopene (LycoRed) 2 mg per day (*n* = 20 women), for 6 months; Control (*n* = 21) hormone replacement therapy (HRT) estradiol valerate 2 mg and norethisterone acetate 1 mg	LycoRed showed beneficial effects on serum lipids and markers of oxidative stress that were comparable to HRT. LycoRed: ↓TC(24.2%); ↓LDL(14.9%); ↑HDL (26.1%), ↓MDA, ↑GSH	[130]
Paran et al., 2009 (Israel)	Blood pressure	Crossover, RCT	50 hypertensive patients with hypertension between 46 and 66 years old; SBP 140–159 mmHg DBP 90–99 mmHg	Encapsulated tomato extract (Lyc-O-Mato^®^) 250 mg, containing 15 mg per day (*n* = 26 men; *n* = 24 women) for 6 weeks; Control (26 men; *n* = 24 women) placebo capsule with soya oil and normal diet for 6 weeks	Tomato extract containing 15 mg of lycopene favorable reduction of SBP from 145.8 to 132.2 mmHg and 140.4 to 128.7 mmHg and DBP 82.1 to 77.9 mmHg. Serum lycopene content increased from 0.30 μmol/L	[107]
Kim et al., 2011 (Korea)	Endothelial function	Parallel, RCT, double-blind	126 healthy men aged 22–57 years	Lycopene in the form (Lyc-O-Mato^®^) 6 mg per day (*n* = 41 men); 15 mg per day (*n* = 37 men) for 8 weeks; Control placebo capsule with soya oil and normal diet for 8 weeks	Supplementation with lycopene in the amount of 15 mg/day for 8 weeks in the group of tested men had a positive effect on endothelial function. Increased the RH-PAT index by 23%. Decreased oxidative DNA damage and increased plasma SOD activity. Moreover, it lowered SBP and the level of: hs-CRP, sICAM-1 and sVCAM-1.	[106]
Burton-Freeman et al., 2012 (USA)	Oxidative stress	RCT, Crossover	25 healthy patients age 27 ± 8 years	Randomly selected patients received (*n* = 12 men and *n* = 13 women): I: 85 g tomato paste/day, II: a diet without the participation of tomatoes (control) by 360 min	Lycopene significantly attenuated high-fat meal ↓LDL oxidation and ↓interleukin-6 a proinflammatory cytokine and a proinflammatory cytokine and inflammation marker	[131]
Xaplanteris et al., 2012 (Greece)	Endothelial function	RCT, Crossover	19 healthy patients age 39 ± 13 years	Patients (*n* = 8 men, *n* = 11 women) received: I: 70 g of tomato paste containing 33.3 mg of lycopene in their diet, II: diet without tomato paste: control for two weeks every day	Tomato paste supplementation increased FMD compared with the control period. It improves the functions of the endothelium. Moreover, it lowers plasma lipid peroxides (TOS)	[132]
Abete et al., 2013 (Spain)	Oxidative stress markers	RCT, Double-blind, Crossover	30 healthy patients aged 39 ± 6 years	Randomly selected patients received (*n* = 9 men, *n* = 21 women): I: tomato sauces 160 g/day containing (27.2 mg lycopene/day), II: commercial tomato sauce 160 g/day with a reduced content of lycopene (12.3 mg lycopene/day) for 10 weeks	The consumption of tomato sauce with a higher concentration of lycopene (27 mg/day) caused a decrease in oxidized LDL-cholesterol levels	[133]
Grajendragadkar et al., 2014 (UK)	Vascular function	Parallel, RCT, double-blind	72 patients, including 36 with cardiovascular disease and 36 healthy volunteers aged 30–80 years	Lycopene 7 mg per day patients healthy (*n* = 23 men, *n* = 1 women) for 2 months; Control (*n* = 10 men, *n* = 2 women) placebo capsule CVD patients (*n* = 15 men, *n* = 9 women): 7 mg per day lycopene, Control (*n* = 10 men, *n* = 2 women) placebo capsule by 8 weeks	Lycopene supplementation in CVD patients improved endothelium-dependent vasodilatation (EDV) by 53%. EDV values were close to the baseline values of healthy patients, indicating an improvement in endothelial function by lycopene. It also caused a slight reduction in SB pressure in patients with CVD by 2.9 mmHg	[112]
Tsitsimpikou et al., 2014 (Greece)	Metabolic syndrome Endothelial function Lipid profile	Parallel, RCT	27 patients with metabolic syndrome aged 53 ± 10 years	Randomly selected patients received: I: tomato juice 100 mL 4 times a week over a period of two months (*n* = 13 men, *n* = 2 women), II: water: control group (*n* = 11 men, *n* = 1 women)	Tomato juice: ↓LDL, ↑HDL. In addition, it lowered the markers of inflammation TNF-α and IL-6. Endothelial function and insulin resistance improved also improved as a result of consuming tomato juice	[134]
Ghavipour et al., 2015 (Iran)	Oxidative stress	RCT	64 overweight and obese female patients (BMI ¼ 25 kg/m^2^ or higher) aged between 20 and 30 years	Female students received: I: tomato juice 330 mL/day containing (37.0 mg/day lycopene) (*n* = 32), II: control: water (*n* = 28) for 20 days	Tomato juice consumption significantly: ↑TAC, ↑erythrocyte SOD, ↑CAT and ↑GPx of plasma and ↓MDA of serum compared with the control group after 20 days	[135]
Deplanque et al., 2016 (France)	Lipid profile	Parallel, RTC	145 healthy patients aged 17–70 years	Patients divided into two groups, the first (75 patients) taking CRTE capsules containing 15 mg/day lycopene and the second (70 patients) taking placebo capsules without lycopene for 2 weeks	Supplementation of CRTE (tomato extract containing 15 mg of lycopene) for 2 weeks increased the plasma level of lycopene and improved the response of oxidized LDL to a high-fat meal in healthy, normal-weight patients. It also had a positive effect on blood glucose, insulin and TG levels	[136]
Valderas-Martinez et al., 2016 (Spain)	Atherosclerosis	RCT, Crossover	40 healthy patients aged 28 ± 11 years	Randomly selected patients received (*n* = 19 men, *n* = 21 women): I: Raw Tomato (RT): 7 g of tomato/kg of body weight, II: tomato sauce (TS): 3.5 g of tomato sauce/kg of body weight; III: tomato sauce with olive oil (TSOO): 3.5 g of tomato sauce with refined olive oil/kg of BW; IV: control: 0.25 g of sugar dissolved in water/kg of BW for 14 weeks	The three groups of products used: RT, TS and TSOO among the examined patients resulted in: ↓TC, ↓LDL, ↑HDL↓. In the assessment of inflammatory markers, the products (TR, TS, TSOO) caused: ↓MCP-1. Whereas, TR and TOOO: ↓IL18 and TOOO: ↓IL6, ↓VCAM-1	[137]
Colmán Martínez et al., 2017 (Spain)	Inflammatory biomarkers Atherosclerosis	RCT, Crossover	28 patients (men) at high risk of cardiovascular disease age 69 ± 3 years	The patients randomly drank in the studies: I: 200 mL (low dose) tomato juice, II: 400 mL (high dose) tomato juice, III: control: water for 4 weeks every day	The use of alternating doses of juices resulted in: ↓ICAM-1, ↓VCAM-1, ↓IL8. A reduction was observed after the consumption of tomato juice (low dose): chemokine CXCL10, CRP, and IFN. However, these changes were not noticed after consuming the juice at a higher dose	[138]
Petyaev et al. 2018 (Russia)	Lipid profile	RCT	142 patients with coronary artery disease aged 45 to 73 years	Patients divided into two groups. One group took lacto-lycopene (Nestle Inc.) (*n* = 68), the other group took microencapsulated GA lycopene (Lycotec Ltd.) (*n* = 74 patients) at 7 mg/day (1 capsule) for 4 weeks	The Ga lycopene (lycosome) supplement caused an increase in serum lycopene concentration in patients compared to lacto-lycopene. At the end of the study, it also lowered the oxidized LDL levels by five times. Such an effect was not observed in patients treated with lacto-lycopene. Ga lycopen also caused an increase in tissue oxygenation and flow-mediated dilation by the end of the observational period	[120]
Wolak et al., 2019 (Israel)	Blood pressure	Parallel, RCT	61 hypertensive patients aged 35–60 years	Patients divided into 5 groups: I receiving the tomato nutrient complex, (TNC) containing 5 mg of lycopene (*n* = 12 patients), II receiving TNC containing 15 mg of lycopene (*n* = 12 patients), III receiving TNC containing 30 mg of lycopene (*n* = 13 patients), IV taking synthetic lycopene in the amount of 15 mg (*n* = 12 patients), V taking a placebo—lycopene-free capsules (soybean oil) (*n* = 12 patients) administered once daily for 8 weeks	Supplementation with TNC containing lycopene 15 (from 137.4 mmHg to 127.2 mmHg) and 30 mg (from 136.4 mmHg to 130 mmHg) caused a decrease in SBP. On the other hand, TNC (lycopene 5 mg) and synthetic lycopene 15 mg had no effect on the decrease of SBP. In the case of diastolic blood pressure (DBP), a TNC containing 15 mg of lycopene had a beneficial effect on its reduction (from 83.8 to 78.6)	[12]

Abbreviation, Reference to the abbreviation: RCT, Randomised controlled trial; TC, Total cholesterol; LDL, Low density lipoprotein; HDL, High density lipoprotein; MDA, Malondialdehyde; GSH, Glutathione; SBP, Systolic blood pressure; DBP, Diastolic blood pressure; RH-PAT, Reactive hyperemia peripheral arterial tonometry; SOD, Superoxide dismutase, hs-CRP, high-sensitivity C-reactive protein; sICAM-1, Soluble inter-cellular adhesion molecule-1; sVCAM-1, Soluble vascular-cellular Adhesion Molecule-1; FMD, Flow-mediated dilatation; CVD, cardiovascular diseases; BMI, Body mass index; TAC, Total antioxidant capacity; CAT, Catalase; GPx, Glutathione peroxidase; CRTE, Carotenoid-rich tomato extract; TG, Triglyceride; TSOO, Tomato sauce with refined olive oil; BW, Body weight; CRP, C-reactive protein; IFN, Interferon. ↑, increase; ↓, decrease.

**Table 3 ijms-23-01957-t003:** Selected studies on animals administered lycopene to determine its cardioprotective protection.

Tested Animal	Lycopene Dose	Effective Operation	Reference
Male Sprague-Dawley (SD) rats injected intraperitoneally with LPS at a dose of 1000 ng/kg body weight/rat to induce infection and inflammation triggered hypertriglyceridemia	(10 mg/kg body weight/day) lycopene was dissolved in corn oil and administered to rats by intra-gastric intubation	Lycopene decreased LPS-induced oxidative stress as well as the expression and level of inflammatory mediators in the plasma. Lycopene treatment down-regulated hepatocyte PCSK-9 expression via down-regulation of HNF-1α.	[139]
Male Wistar rats injected with angiotensin II (Ang II) (0.3 mg/kg/day)	Lycopene was incorporated into drinking water (the Redivivo^®^ product) 10 mg/kg/day	Lycopene lowered angiotensin II-induced hypertension. Prevented the increase in left ventricular weight and therefore the LVH index. Prevented cardiomyocyte hypertrophy and myocardial fibrosis.	[66]
Male Kunming mice ATR (atrazine)-induced cardiac inflammation (50 mg/kg, 200 mg/kg)	5 mg/kg body weight/day	Supplementary LYC significantly prevented ATR-induced environmental cardiology and alleviated cardiac injury via modulating the NO and NO-generating systems and blocking the TRAF6-NF-κB. Lycopene was lowering pro-inflammatory TNF-α, IL-6, COX2 and IL-1β.	[64]
Male BALA/c mice (18–25 g; 10–12 weeks) Left anterior descending coronary artery ligated	10 mg/kg body weight/day	Lycopene significantly reduced the mRNA expression of TNF-α and IL-1β in infarcted myocardium. Decreased expression of caspase-3, -8 and -9. In addition, lycopene decreased the nuclear localization of nuclear NF-κB p65 and markedly suppressed NF-κB activation.	[45]
Male Wistar albino rats Isoproterenol (85 mg/kg) has been used to induce myocardial infarction in 29 and 30 day (two days)	(0.5, 1.0 and 1.5 mg/kg body weight/day) for 30 days	Initial treatment with lycopene significantly attenuated ISP-induced cardiac dysfunction as evidenced by improved SAP, DAP, MAP, HR, (±) LVdP/dt and reduced LVEDP. Lycopene also prevented the depletion of antioxidants (SOD, CAT, GSHPx and GSH) and inhibited lipid peroxidation and MDA formation in the heart.	[140]

Abbreviation, Reference to the abbreviation: LPS, Lycopene in targeting lipopolysaccharide; PCSK-9, Proprotein convertase subtilisin/kexin type-9; HNF-1α, Hepatocyte nuclear factor-1α; LVH, Left ventricular hypertrophy; LYC, Lycopene; NO, Nitric oxide; TRAF6-NF-κB, Tumor necrosis factor receptor-associated factor 6-nuclear factor-κB; TNF-α, Tumor necrosis factor α; IL-6, Interleukin-6; COX2, Cyclooxygenase 2; IL-1β, Interlekuin-1β; ISP, Isoproterenol; SAP, Systolic pressure; DAP, Diastolic pressure; MAP, Mean arterial blood pressure; HR, Heart rate; LVEDP, Left ventricular end-diastolic pressure; SOD, Superoxide dismutase; CAT, Catalase; GSHPx, glutathione peroxidase; GSH, glutatione, MDA, Malonaldialdehyde.

## Data Availability

Not applicable.

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
