# Peer review of "Lycopene in the Prevention of Cardiovascular Diseases"

_ijms, 2022, doi:10.3390/ijms23041957_

Round 1

Reviewer 1 Report

The authors have made a laborious work, reviewing published data concerning the protective potential of Lycopene on the cardiovascular system. The manuscript is well-written and constructed and some minor comments emerged.

  1. Some clinical metanalyses addressing the anti-inflammatory potential of lycopene in terms of CVDs are missing from the manuscript (PMID: 28129549, 33131949, etc). Authors are adviced to update their literature search and include the aforementioned meta-analyses within their review.
  2. The manuscript would benefit from some graphical illustration, representing and summarizing the effects of lycopene on the CV system and the molecular signaling induced by lycopene.
  3. Within their manuscript authors pinpoint the molecular targets of lycopene. However, it is clear that lycopene regulates the same molecular targets as the majority of natural products. For instance ARE transcriptional targets are shared among many natural products (i.e resveratrol, quercetin, hydroxytyrosol etc). It would be interesting if authors could identify a more "specific" molecular fingerprint of lycopene, which could further support its nutraceutical preponderance.

Author Response

Thank you very much for the review for my publication ID IJMS-1508631 "Lycopene in the prevention of cardiovascular diseases"

I have made corrections in line with the Reviewer's comments. I marked red in my work - all that was edited by me. I have included tables of clinical trials and meta-analyzes as well as graphical studies and additional information about lycopene.

Please, re-verify my work.

Thank you very much.

Reviewer 2 Report

  • The authors should include a figure able to improve the readability of the text and outlining the main mechanisms able to evaluate the impact of Lycopene in CVD.
  • The authors should also discuss the role of Lycopene as antioxidant element. They can consider and discuss the papers from Ciccone MM et al. Mediators Inflamm. 2013;2013:782137 and Giordano P et al. Curr Pharm Des. 2012;18(34):5577-89.
  • A table gathering the main findings from CLINICAL studies should be included in this review.

Author Response

Thank you very much for the review for my publication ID IJFST-1508631 "Lycopene in the prevention of cardiovascular diseases"

I have made corrections in line with the Reviewer's comments. I marked red in my work - all that was edited by me. I have included tables of clinical trials and meta-analyzes as well as graphic studies and information on the antioxidant properties of lycopene.

Please, re-verify my work.

Thank you very much.